# OpenReview forum: "Unlearning That Lasts: Utility-Preserving, Robust, and Almost Irreversible Forgetting in LLMs"
_ICLR.cc/2026/Conference — Submitted to ICLR 2026_

### Official Review · Reviewer_tjmy · 2025-10-20

**Soundness:** 2
**Presentation:** 1
**Contribution:** 2
**Rating:** 2
**Confidence:** 5

**Summary:**

This paper introduces JensUn, a new LLM unlearning method that uses Jensen–Shannon Divergence to achieve a better balance between forgetting targeted knowledge and preserving overall model utility. The authors show that JensUn is more stable and resistant to benign relearning, meaning the forgotten information is less likely to re-emerge after additional fine-tuning. To address weaknesses in current evaluation practices, they propose a stronger assessment pipeline that uses LLM-based semantic judging instead of ROUGE, tests multiple paraphrased query versions, and reports worst-case performance. They also introduce a new dataset, LKF (Lesser Known Facts), for more realistic unlearning scenarios. Overall, the paper argues that many existing unlearning methods appear effective only under weak evaluations, while JensUn demonstrates more durable and robust forgetting.

**Strengths:**

1. The motivation for introducing a new unlearning method is strong.

2. The paper proposes a new evaluation approach using an LLM as a judge.

3. The experimental section is comprehensive and includes various evaluation metrics.

**Weaknesses:**

1. I do not see the novelty of the proposed unlearning method. The paper introduces JensUn, which uses JSD for the forget and retain losses; however, this design choice appears to be based only on preliminary experiments, and there is no theoretical justification provided.

2. The writing of the paper can be significantly improved. For example, the Introduction lacks a concluding paragraph, making it difficult for readers to clearly grasp the main contributions. Additionally, based on the writing, I assumed that the LKF dataset is one of the primary contributions, yet it is only presented as a subsection in the paper, with significantly less description compared to the introduction of evaluation metrics, which are not core technical contributions.

**Questions:**

Could you provide theoretical analysis to explain why JSD is a better choice for the unlearning objective?

---

> ### Author Response · Authors · 2025-11-18
> **Rebuttal**
>
> We thank the reviewer for their comments. However, we disagree with the reviewer in several key points, which we will outline below. We have uploaded a revised version (changes in red).
>
> 1. **“no novelty of method, and lack of theoretical justification”, “Could you provide theoretical analysis to explain why JSD is a better choice for the unlearning objective?”**
> We would like to point the reviewer to the discussion at the end of Section 3.1 (“Why Jensen-Shannon Divergence?”), and to Appendix E.3 and E.4, where we provided extensive discussion on this. In the revised version, we made those insights more prominent. In particular, we
>     * formulated our analysis of the $\ell_1$-norm of the gradient of the KL-divergence and JSD of the forget loss into Lemma 1 and moved it to the main paper
>     * moved Figure 16 to the main paper (now Figure 2), which provides empirical evidence for the insights of our Lemma
>
>     Lemma 1 shows that at the beginning of unlearning, when the probability $q_k$ of the token of the refusal target is small ($q_k \approx 0$), the KL-divergence gradient norm of the forget loss is maximally large, while the norm of the gradient of the JS-divergence gradient of the forget loss is close to zero. Moreover, for both losses the gradient of the retain loss aiming to preserve the output of the original LLM is zero at initialization.
>     This imbalance at initialization for KL leads to larger changes of the model, which are detrimental to the utility of the LLM, as shown in Figure 2, and from which one cannot recover by further training. For JSD, in contrast, the gradients of forget and retain loss are almost balanced, at initialization and thus lead to changes of the model which enforce unlearning, but at the same time maintain the utility of the LLM, please see Figure 2, which shows that the win-rate (measure of model utility relative to the original model) stays close to the optimal value of 0.5 throughout the unlearning process, whereas for KL we see a sharp drop of utility at the initial unlearning stage. Both KL and JSD achieve low forget accuracy (1% and 0%) but JSD yields the better utility-forget-accuracy trade-off.
>     We would like to emphasize that all of this analysis and discussion was already present in the submitted version. However, we agree that the paper benefits from making it more prominent.
>
>
>
>
>
> 2. **“Introduction lacks a concluding paragraph”**
> We added a contributions paragraph after the introduction in the revised version.
>
> 3. **“LKF does not have enough visibility”**
> We agree that LKF is a significant contribution of this work, and deserves more visibility. In the revised version, we have added more explanations, and samples from the dataset to the main paper (Section 4.4 and Figure 4). Due to the page limit, the details on the creation process are deferred to Appendix A.
>
> 4. **Neglection of experimental results**
> In their summary, the reviewer mentions that JensUn is more stable and resistant to benign relearning, which we agree to. However, we find it unfortunate that this is not reflected in the reviewers strength section. Similarly, the reviewer describes that JensUn better balances forgetting and overall utility, which we also agree to. Again, this is not reflected in the strengths that the reviewer mentions
>
> In summary, we think that the weaknesses mentioned by the reviewer (lack of theoretical justification - we provide extensive justification, and poor writing - reviewer jjRj explicitly appreciates the clear writing, and we added the requested conclusions paragraph) do not justify the score given by the reviewer, especially since our strong experimental results seem not to play a role in their assessment. Given our clarifications, which we hope solve the raised concerns and some apparent misunderstandings, we would like to ask the reviewer to reconsider their score. We are happy to engage in further discussion and resolve remaining doubts.

---

### Official Review · Reviewer_roR2 · 2025-11-01

**Soundness:** 3
**Presentation:** 3
**Contribution:** 2
**Rating:** 4
**Confidence:** 3

**Summary:**

This paper proposes JensUn, an LLM unlearning method that replaces the existing loss functions with Jensen-Shannon divergence (JSD). Specifically, on the forget set, the authors construct a target answer, such as refusal, and minimize the JSD between the model's output distribution with the target answer distribution. On the retain set, the authors minimize the JSD with the reference model's output. Moreover, the authors propose several improvements in evaluating an unlearned model, such as using the worst-case performance across multiple paraphrases of the question, and adding retain set examples in context. Experiments on two datasets show that the proposed method outperforms strong baselines.

**Strengths:**

1. The proposed method is simple, and experiments show that it outperforms the strong baselines, achieving a better trade-off between unlearn and retain performance.
2. The authors propose several improvements for evaluating unlearned models. These metrics contribute to a more robust unlearning evaluation.

**Weaknesses:**

1. My main concern is the lack of a comprehensive ablation study. Particularly, the authors do not show that it is the JSD that improves the performance. For example, since the authors construct a refusal target for the forget data, what's the performance of applying other loss functions (maximize loglikelihood, preference optimization, or minimize KL) on the same refusal target? On the retain set, is JSD necessary, or is KL enough?

2. Although the proposed metrics contribute to a more robust evaluation, these metrics are already introduced and used in prior works. For example, LLM-as-judge to evaluate the answer equivalence is widely used. Particularly for unlearning, paraphrases have been used [1], and worst-case performance under adversarial attacks has also been explored [2-4].

[1] Wang et al., Towards Effective Evaluations and Comparisons for LLM Unlearning Methods.
[2] Lynch et al., Eight Methods to Evaluate Robust Unlearning in LLMs.
[3] Schwinn et al., Soft Prompt Threats: Attacking Safety Alignment and Unlearning in Open-Source LLMs through the Embedding Space.
[4] Liu et al., Revisiting Who's Harry Potter: Towards Targeted Unlearning from a Causal Intervention Perspective.

**Questions:**

1. Can the authors provide more explanations for why JSD is better than other loss functions? For example, how's the gradient different and more stable than other methods?

---

> ### Author Response · Authors · 2025-11-18
> **Rebuttal 1/3**
>
> We thank the reviewer for reading our paper and for providing helpful comments. We are glad that the reviewer acknowledged that our method “outperforms the strong baselines, achieving a better trade-off between unlearn and retain performance”, and also appreciates that our evaluation framework allows for a “more robust unlearning evaluation”. We have uploaded a revised version (changes in red). Here, we address the reviewers concerns:
>
> 1. **“what's the performance of applying other loss functions (maximize loglikelihood, preference optimization, or minimize KL) on the same refusal target?”**
> We would like to clarify that the log-likelihood (or Cross-entropy) w.r.t. a target string is __equivalent__ to the KL-divergence with a target string. We evaluate the KL-divergence with our target string already (see Table 1), and report improved performance with JensUn. We provide extensive discussion on why this is the case (see Section 3.1, Appendix E.3 and E.4), including theoretical results that show that the KL-divergence (=CE with target string) leads to unbalanced gradients at the beginning of training, which we confirm experimentally in Figure 2 (previously Figure 16). The results for KL in Table 1 also indicate that for this method, achieving low forget accuracy is only possible by compromising the model utility and retain performance - different to JensUn. Additionally, we evaluated preference optimization where the winner-sample is our refusal string “no idea”, and the loser-sample is the true forget answer, and found that it did not perform well (JensUn Numbers are slightly different to main paper because we had to use a different judge):
> | method  |          forget (J_W)   | retain (J_avg)   |     MMLU |     REP |     WR |
> |:----------------|:---------------------|:-----------------|---------:|--------:|-------:|
> | JensUn          |         0.0                 | 54.7            | 59.1 | 581  |   0.52 |
> | Pref-no-idea         |         45.0                 | 62.1            | 58.1 | 468 |   0.50 |
>
> 2. **“On the retain set, is JSD necessary, or is KL enough?”**
> The importance of JSD comes from its use as forget loss not as retain loss. So we expect that using the KL instead of the JSD should not bring big changes, because if the divergence is computed with respect to the base-model, the retain loss and, in particular its gradient, will be zero at the beginning of unlearning for both losses. The important insight of our paper is that the gradient of JSD for the forget loss is also close to zero at initialization, whereas the norm of the gradient of the KL-divergence is maximal. Please see the answer below to the question why the JSD loss is better than other losses.
> We can confirm this with an ablation (like above, JensUn numbers are slightly different to main paper because we had to use a different judge). As expected, there is essentially no difference, and both variants bring strong results:
>
>
> | forget loss   | retain loss   |          forget (J_W)   | retain (J_avg)   |     MMLU |     REP |     WR |
> |:----------------|:----------------|:---------------------|:-----------------|---------:|--------:|-------:|
> | JSD          | JSD          |         0.0                 | 54.7            | 0.590682 | 580.852 |   0.52 |
> | JSD          | KL              |         0.0                 | 54.2            | 0.590682 | 580.376 |   0.51 |

---

> ### Author Response · Authors · 2025-11-18
> **Rebuttal 2/3**
>
> 3. **“these metrics are already introduced and used in prior works.”**
> We agree that parts of our evaluation framework have been proposed in prior works, as acknowledged throughout the paper (e.g. lines 213, 222, 236, 238, 240, 250, 284). In fact, it was motivated by previous works like [5], which, amongst other things, argue that paraphrases should be used for evaluation. We also mention other works like [8] that used paraphrases, and show that our evaluation framework is stronger. We also don’t claim that we are the first to use an LLM-judge - we cite several works from other fields - but we show how for unlearning it improves over the brittle ROUGE scores that is used in several benchmarks. We would like to clarify that [1], which was mentioned by the reviewer, evaluated the paraphrased _answers_ of TOFU. As shown by [5], this is still brittle to paraphrases in the questions - which we fix with our evaluation framework. Regarding worst-case performance, we agree that worst-case performance under adversarial attacks has been evaluated before [2,3,4,8]. However, we also show that our worst-case over benign paraphrases of questions, that is directly motivated by the findings of [5], is effective for unlearning evaluation when combined with a judge. In particular, it is stronger than e.g. the adversarial setup in [8]. Finally, we would like to mention that very recent works about unlearning still use the ROUGE score, or don’t report worst-case evaluations over paraphrases (e.g. [6,7] ). Establishing an effective, unified evaluation protocol, which we do, is therefore needed for the community. We have included references [1,2,3,4] in the revised version.
>
>
> 4. **“Can the authors provide more explanations for why JSD is better than other loss functions? For example, how's the gradient different and more stable than other methods?”**
> We would like to point the reviewer to the discussion at the end of Section 3.1 (“Why Jensen-Shannon Divergence?”), and to Appendix E.3 and E.4, where we provided extensive discussion on this. In the revised version, we made those insights more prominent. In particular, we
>     * formulated our analysis of the $\ell_1$-norm of the gradient of the KL-divergence and JSD of the forget loss into Lemma 1 and moved it to the main paper
>     * moved Figure 16 to the main paper (now Figure 2), which provides empirical evidence for the insights of our Lemma
>
>     Lemma 1 shows that at the beginning of unlearning, when the probability $q_k$ of the token of the refusal target is small ($q_k \approx 0$), the KL-divergence gradient norm of the forget loss is maximally large, while the norm of the gradient of the JS-divergence gradient of the forget loss is close to zero. Moreover, for both losses the gradient of the retain loss aiming to preserve the output of the original LLM is zero at initialization.
>     This imbalance at initialization for KL leads to larger changes of the model, which are detrimental to the utility of the LLM, as shown in Figure 2, and from which one cannot recover by further training. For JSD, in contrast, the gradients of forget and retain loss are almost balanced, at initialization and thus lead to changes of the model which enforce unlearning, but at the same time maintain the utility of the LLM, please see Figure 2, which shows that the win-rate (measure of model utility relative to the original model) stays close to the optimal value of 0.5 throughout the unlearning process, whereas for KL we see a sharp drop of utility at the initial unlearning stage. Both KL and JSD achieve low forget accuracy (1% and 0%) but JSD yields the better utility-forget-accuracy trade-off.
>     We would like to emphasize that all of this analysis and discussion was already present in the submitted version. However, we agree that the paper benefits from making it more prominent.

---

> ### Author Response · Authors · 2025-11-18
> **Rebuttal 3/3**
>
> 5. **Robustness to relearning**:
> We think the reviewer overlooked one of the main contributions of JSD: Its ability to train for longer, and its robustness to relearning (Table 3 in main paper, more in Appendix). Unlearning is arguably pointless, if it can be undone by fine-tuning on a set that is unrelated to the forget set. Thus, being robust to relearning is extremely important, and JensUn shines in this setup, clearly outperforming other methods. We would appreciate it if this were included in the reviewers assessment.
>
> In light of these clarifications, which address the raised concerns and some apparent misunderstanding, we would like to ask the reviewer to reconsider their score. We are happy to provide further clarifications.
>
> [1] Wang et al., Towards Effective Evaluations and Comparisons for LLM Unlearning Methods.
>
> [2] Lynch et al., Eight Methods to Evaluate Robust Unlearning in LLMs.
>
> [3] Schwinn et al., Soft Prompt Threats: Attacking Safety Alignment and Unlearning in Open-Source LLMs through the Embedding Space.
>
> [4] Liu et al., Revisiting Who's Harry Potter: Towards Targeted Unlearning from a Causal Intervention Perspective
>
> [5] Pratiksha Thaker, Shengyuan Hu, Neil Kale, Yash Maurya, Zhiwei Steven Wu, and Virginia Smith. Position: Llm unlearning benchmarks are weak measures of progress. In SaTML, 2025
>
> [6] Krishnan et al, Not All Data Are Unlearned Equally, COLM 2025
>
> [7] Bushipaka et al,  Standard vs. Modular Sampling: Best Practices for Reliable LLM Unlearning, https://arxiv.org/pdf/2509.05316
>
> [8] Zhuoran Jin et al. Rwku: Benchmarking real-world knowledge unlearning for large language
> models. In NeurIPS Datasets and Benchmarks Track, 2024

---

### Official Review · Reviewer_jjRj · 2025-11-01

**Soundness:** 3
**Presentation:** 2
**Contribution:** 2
**Rating:** 4
**Confidence:** 3

**Summary:**

This paper proposes a new unlearning loss to improve the utility preserving in LLM unlearning, which leverages Jensen-Shannon Divergence, that optimizes the unlearning LLM to match the distribution of refusing words like "No idea" on forget set. It further introduces a new unlearning evaluation dataset LKF focusing on lesser-known facts in web data. The experiments show that it improves the utility preservation while having matchable forget performance.

**Strengths:**

* The idea is simple and straightforward to using the Jenson-Shannon Divergence as a distribution match loss for LLM unlearning purpose.
* The writing is clear and easy to follow.
* LKF dataset on lesser-known facts may be a good contribution to the field.

**Weaknesses:**

* Unconvincing performance gain. While the proposed JS divergence loss is easy to understand, I don't have the intuition that it is better than other unlearning forget loss, like those optimizing the LLM to output refusing words like "I don't know" on forget data. The JS divergence seems yet another loss pushing the LLM to output refusing words. And the performance in Table 1/2 also looks contradicted to me, where the utility preserving seems not so different from other KL divergence method, but much achieves higher forget performance compared to other methods like NPO/SimNPO.
* Inadequate ablation on retain loss selection. The retain loss in the proposed framework is also the JSD loss. However, there is no ablation study on how it performs when combining with other LLM forget loss.

**Questions:**

* I feel confused about some of the performance result in Table 1. The retain accuracy for SimNPO is 84.2, largely outperforms other baselines and even the original model, why is this case?
* From the example responses in Figure 15, JensUn unlearn response is repeated No idea, which is a bad-quality response, and it seems to contradict with the repetitive performance in main paper.

---

> ### Author Response · Authors · 2025-11-18
> **Rebuttal 1/2**
>
> We thank the reviewer for reading our paper and for their feedback. We are glad that the reviewer likes our idea, finds our writing clear, and appreciates the LKF benchmark as a valuable contribution to the field. We have uploaded a revised version (changes in red). We would like to clarify several points raised by the reviewer, and provide additional comments on our other contributions:
>
> 1. **“I don't have the intuition that JSD is better than other unlearning forget loss”**
> We would like to point the reviewer to the discussion at the end of Section 3.1 (“Why Jensen-Shannon Divergence?”), and to Appendix E.3 and E.4, where we provided extensive discussion on this. In the revised version, we made those insights more prominent. In particular, we
>     * formulated our analysis of the $\ell_1$-norm of the gradient of the KL-divergence and JSD of the forget loss into Lemma 1 and moved it to the main paper
>     * moved Figure 16 to the main paper (now Figure 2), which provides empirical evidence for the insights of our Lemma
>
>     Lemma 1 shows that at the beginning of unlearning, when the probability $q_k$ of the token of the refusal target is small ($q_k \approx 0$), the KL-divergence gradient norm of the forget loss is maximally large, while the norm of the gradient of the JS-divergence gradient of the forget loss is close to zero. Moreover, for both losses the gradient of the retain loss aiming to preserve the output of the original LLM is zero at initialization.
>     This imbalance at initialization for KL leads to larger changes of the model, which are detrimental to the utility of the LLM, as shown in Figure 2, and from which one cannot recover by further training. For JSD, in contrast, the gradients of forget and retain loss are almost balanced, at initialization and thus lead to changes of the model which enforce unlearning, but at the same time maintain the utility of the LLM, please see Figure 2, which shows that the win-rate (measure of model utility relative to the original model) stays close to the optimal value of 0.5 throughout the unlearning process, whereas for KL we see a sharp drop of utility at the initial unlearning stage. Both KL and JSD achieve low forget accuracy (1% and 0%) but JSD yields the better utility-forget-accuracy trade-off.
>     We would like to emphasize that all of this analysis and discussion was already present in the submitted version. However, we agree that the paper benefits from making it more prominent.
>
>
>
> 2. **“performance in Table 1/2 also looks contradicted to me, where the utility preserving seems not so different from other KL divergence method, but much achieves higher forget performance compared to other methods like NPO/SimNPO”**
> It is important to emphasize that there is a tradeoff between forget performance and other metrics like retain accuracy, utility, win rate (WR), etc. By tuning the loss coefficients or the learning rate, for each method, we can obtain stronger forgetting by compromising the performance on nominal tasks, and vice versa. The values in Table 1 and 2 thus are points on this tradeoff curve. When looking at the curves directly (Figure 3), it becomes obvious that JensUn outperforms the other methods clearly, and is pareto-optimal. Further, in Table 1, MMLU performance is indeed similar for several methods, but this does not capture the models utility, as explained in Section 4.3: The other methods that achieve good forget performance (KL-Div, GradAscent, GradDiff, NPO), are clearly outperformed in terms of win-rate, indicating that they suffered a degradation in nominal utility, which is not captured by MMLU, since models can achieve high MMLU performance even when they do not output sensible/fluent responses anymore.
>   In summary, JensUn is basically optimal on LKF (Table 1). It achieves perfect forget performance (forget accuracy is zero), maintains the performance of the original model on the retain set and MMLU, and in particular it has a win-rate of 0.47 that means the LLM judge sees almost no difference between the output of the original LLM and our JensUn-finetuned version. The only other method which has a similar high win rate of 0.47 is RMU which however has still a high forget accuracy of 19\% and suffers a 3\% loss on MMLU. Similar DPO has a worse win-rate of 0.42 but also a much worse unlearning performance with a forget accuracy of 32\%.

---

> ### Author Response · Authors · 2025-11-18
> **Rebuttal 2/2**
>
> 3. **“The retain accuracy for SimNPO is 84.2, largely outperforms other baselines and even the original model, why is this case?”**
> The reason why some methods achieve high retain performance (even higher than the base model), is that those methods explicitly minimize some loss on the retain set (typically CE), and not the divergence to the base model. If retain and forget loss cannot be perfectly balanced, this can lead to overfitting on the retain set (which is what we observe here). Typically, this comes at the expense of loss in utility, or bad forget performance. It is a deliberate design choice of our method to use the JSD to the outputs of the base model, in order to preserve utility beyond the retain set as much as possible. In fact, we think that the goal of unlearning is to remove the information in the forget set while _preserving_ the same behaviour as the pre-unlearning model on any other set, which our method aims for.
>
> 4. **“From the example responses in Figure 15, JensUn unlearn response is repeated No idea, which is a bad-quality response, and it seems to contradict with the repetitive performance in main paper.”**
> These repetitive answers only appear for forget questions, and we are ok with a repetitive answer, as long as the correct answer is not revealed by the unlearnt model. On questions unrelated to the forget queries, where we want to preserve high quality and avoid repetitions, JensUn performs extremely well (as can be seen from the repetitiveness measure and the win rate, and from Figures 22,23,24).
>
> 5. **“there is no ablation study on how it performs when combining JSD as retain loss with other LLM forget losses”**
> We want to emphasize that we see no obvious reason why the JSD should bring advantages for the **retain loss** over other divergences to the base model. The core of our analysis is having the JSD as **forget loss**, as it mitigates the problem of other losses (such as KL), which have large gradients at the beginning of training. Having some other forget loss, and the JSD as retain loss is not motivated by this - JSD as retain cannot fix the problems of the other forget losses. On the other hand, we could replace the JSD as retain loss in our method with the KL, and should not expect large differences - because the JSD as forget has small gradients initially, and the KL as retain also. We can confirm this with an ablation (numbers are slightly different to the main paper because we had to use a different judge). As expected, there is essentially no difference, and both variants bring strong results:
> | forget loss   | retain loss   |          forget (J_W)   | retain (J_avg)   |     MMLU |     REP |     WR |
> |:----------------|:----------------|:---------------------|:-----------------|---------:|--------:|-------:|
> | JensUn          | JensUn          |         0.0                 | 54.7            | 59.1 | 581 |   0.52 |
> | JensUn          | KL              |         0.0                 | 54.2            | 59.1 | 580 |   0.51 |
>
>
> 6. **robustness to relearning:**
> We think the reviewer overlooked one of the main contributions of JensUn: Its ability to train for longer, and its robustness to relearning (Table 3 in main paper, more in Appendix). Unlearning is arguably pointless, if it can be undone by fine-tuning on a set that is unrelated to the forget set. Thus, being robust to relearning is extremely important, and JensUn shines in this setup, clearly outperforming other methods.
>
> 7. **Eval framework:**
> We also think that the reviewer might have overlooked our improved evaluation framework, which fixes flaws that existed in previous evaluations.
>
> In light of these clarifications, which solve the raised concerns and some apparent misunderstandings, we would like to ask the reviewer to reconsider their score. We are happy to provide further clarifications.

---

### Author Response · Authors · 2025-12-03
**Summary of Rebuttal Period**

Dear AC,

in order to facilitate the assessement of our paper after the rebuttal period, we provide a very brief explanation of how we addressed the central concerns of each reviewer.

**Reviewer jjRj**
* **Concern:** The reviewer questioned the intuition behind why JSD is superior to other unlearning losses (like KL) and perceived contradictions in the utility-forgetting trade-off compared to baselines like SimNPO.
* **Resolution:** We believe the reviewer overlooked our "Why JSD?" paragraph, and especially the theoretical analysis in the Appendix. We have summarized it into a Lemma in the main paper (Lemma 1), and together with our empirical results (Figure 2), it demonstrates that unlike KL, which suffers from large gradients of the forget loss at initialization that destroy model utility, JSD balances the gradients of forget and retain loss and thus achieves excellent forget performance while maintaining utility. We further clarified that baselines with higher retain accuracy than the base model often overfit the retain set, whereas JensUn achieves a superior, Pareto-optimal trade-off.

**Reviewer roR2**
* **Concern:** This reviewer requested ablations to isolate the benefit of JSD over standard losses (KL, CE, DPO) on the refusal target, questioned the novelty of our evaluation metrics relative to prior work, and asked for explanations why JSD is better than other loss functions.
* **Resolution:** We explained that such experiments are already present in the paper for KL and CE (which are equivalent for a fixed target), provided ablations with DPO, and explained that our theoretical analysis from the paper showed that JSD's specific gradient properties on the forget set are essential for stability, whereas the choice of retain loss is less important (KL could be used equally, which we confirm experimentally). We also clarified that while individual components of our evaluation exist in literature, our unified framework is a novel contribution that resolves specific brittleness in current benchmarks. We highlighted Lemma 1 (previously in Appendix) and our explanations in the "Why JSD?" paragraph, which was likely overlooked by the reviewer.

**Reviewer tjmy**
* **Concern:** The reviewer stated the method lacked theoretical justification and novelty, felt the LKF dataset was not prominent enough in the writing, and asked for a Contributions paragraph.
* **Resolution:** We strongly refuted the lack of theory by highlighting our gradient analysis (now Lemma 1), which was likely overlooked by the reviewer. We also revised the paper to feature the LKF dataset more prominently and added a specific contributions paragraph as requested. Finally, we noted that the reviewer’s scoring did not seem to account for the strong experimental results they acknowledged in their summary.


We believe these responses fully resolve the concerns raised, and kindly refer to the full rebuttal comments for more details.

Best,
Authors

---

### Meta-Review · Area_Chair_21Xx · 2026-01-06

**Summary:**

The reviewers' primary concerns centered on the perceived lack of theoretical justification for the superiority of Jensen-Shannon Divergence (JSD) over existing loss functions like KL-divergence, insufficient ablation studies to isolate the contribution of JSD, and questions about the novelty of the proposed evaluation framework relative to prior work. I feel that the adoption of JSD may not be an enough contribution for accepting this paper, given that the proposed evaluation framework is a totally separate part and not clearly validated to be a sufficiently original contribution.

**Reviewer Concerns:**

The authors' rebuttal effectively addressed several key concerns. The authors moved theoretical gradient analysis into the main paper as Lemma 1, provided ablation studies comparing JSD with KL on the retain loss, added a contributions section, and increased the prominence of the LKF dataset. However, some concerns remain outstanding, particularly regarding whether the fundamental novelty of using JSD is fully established beyond gradient stability, and whether the evaluation framework constitutes a sufficiently original contribution given existing literature on paraphrasing and LLM-as-judge metrics.

**Reviewer Scores:**

Reviewer jjRj, might be satisfied with the clarification on gradient dynamics and the Pareto-optimal trade-off, and might increase the score to weak accept.

Reviewer roR2, would likely increase their score to weak accept, as the specific requests for ablations and explanations were responded by the authors.

Reviewer tjmy, who gave a reject score with high confidence and emphasized the limited novelty of the paper, would probably keep the reject score.

---

### Decision · Program_Chairs · 2026-01-26

Reject